# Hepatitis E Virus (HEV) Infection in Hemodialysis Patients: A Multicenter Epidemiological Cohort Study in North-Eastern Greece

**DOI:** 10.3390/pathogens12050667

**Published:** 2023-04-30

**Authors:** Dionysios Kogias, Aikaterini Skeva, Andreas Smyrlis, Efthymia Mourvati, Konstantinia Kantartzi, Gioulia Romanidou, Maria Kalientzidou, Vasiliki Rekari, Eleni Konstantinidou, Parthena Kiorteve, Ioannis Paroglou, Vasileios Papadopoulos, Theocharis Konstantinidis, Maria Panopoulou, Konstantinos Mimidis

**Affiliations:** 1First Department of Internal Medicine, Democritus University of Thrace, 68100 Alexandroupolis, Greece; dionkogi@gmail.com; 2Laboratory of Microbiology, Department of Medicine, Faculty of Health Sciences, Democritus University of Thrace, 68100 Alexandroupolis, Greece; skevakaterina@hotmail.com (A.S.); mpanopou@gmail.com (M.P.); 3Department of Nephrology, Democritus University of Thrace, 68100 Alexandroupolis, Greece; andrsmyrlis@gmail.com (A.S.); drefthm@gmail.com (E.M.); kokan0910@gmail.com (K.K.); 4Nephrology Department, General Hospital “Sismanogleio”, 69100 Komotini, Greece; dr_giouliarom@yahoo.gr; 5Department of Nephrology, General Hospital of Kavala, 65500 Kavala, Greece; merinakali18@gmail.com; 6Blood Transfusion Center, General Hospital of Xanthi, 67100 Xanthi, Greece; vickyrekari@gmail.com; 7Blood Transfusion Center, General Hospital of Drama, 66100 Drama, Greece; helenkonst28@gmail.com; 8Nephrology Department, General Hospital of Drama, 66100 Drama, Greece; kiorteve@yahoo.gr; 9Nephrology Department, General Hospital of Didymoteicho, 68300 Didymoteicho, Greece; john_paroglas@yahoo.gr; 10AKESIOS Dialysis Center, 67100 Xanthi, Greece; vaspapmd@gmail.com; 11Blood Transfusion Center, University General Hospital of Alexandroupolis and Laboratory of Microbiology, Democritus University of Thrace, 68100 Alexandroupolis, Greece; theoxari_ko@yahoo.gr; 12Laboratory for the Study of Gastrointestinal System and Liver, Democritus University of Thrace, 68100 Alexandroupolis, Greece

**Keywords:** Greece, hepatitis E virus, dialysis, hemodialysis, prevalence, seroprevalence, HEV RNA, risk factors, epidemiology

## Abstract

Hepatitis E virus (HEV), a common cause of viral hepatitis in developing countries, is mainly transmitted via the fecal–oral route, but also may be a prevalent hospital-transmitted agent among patients on regular hemodialysis due to parenteral transmission. Previous epidemiological studies among hemodialysis patients in Greece, using different diagnostic techniques, gave conflicting results. Τhe present study aimed to measure the exposure rate of hemodialysis patients of north-eastern Greece to HEV by estimating the overall seroprevalence, and to identify potential risk factors. Serum samples from all patients attending the hemodialysis centers of north-eastern Greece (n = 6) were tested for the presence of anti-HEV IgG antibodies using a modern and sensitive ELISA (Enzyme-linked Immunosorbent Assay) technique (Wantai). In total, 42 out of 405 hemodialysis patients were positive for anti-HEV IgG (10.4%), while all samples were negative for HEV RNA when tested using nested RT-PCR. HEV seropositivity among hemodialysis patients was significantly associated with area of residence and contact with specific animals (pork, deer). No association was found with religion, gender distribution and hemodialysis duration. This study showed an increased seroprevalence of HEV among hemodialysis patients in Greece. Agricultural or livestock occupation and place of residence seem to be independent factors that increase the risk of HEV infection. In conclusion, HEV infection calls for the regular screening of hemodialysis patients regardless of the hemodialysis duration or clinical symptoms.

## 1. Introduction

Hepatitis E virus (HEV) is a common cause of viral hepatitis in developing countries, most commonly transmitted through the fecal–oral route. HEV belongs to the Hepeviridae family of the genus Hepevirus; it is a spherical, non-enveloped, single-stranded, positive sense RNA virus with an approximately 32 to 34 nm diameter [1]. The molecular phylogenetic analysis classifies HEV into four major genotypes (HEV1-4) and 24 sub-types [1,2]. HEV1 and HEV2 have been found only in humans and are often associated with epidemics in developing countries due to poor hygiene and sanitation. In contrast, HEV3 and HEV4 were first isolated from swine, and have the ability to circulate in several animals, including wild boars, and deers, without causing any disease. For this reason, they are referred as zoonotic. These genotypes are prevalent in industrialized countries and are associated with sporadic and clustered cases of hepatitis E [2]. Regarding the recent increases in reported outbreaks of HEV3 and HEV4 through blood transfusions in Europe, many countries consider the screening of blood products for HEV [3,4]. Furthermore, apart from being symptomatic, the virus may cause an asymptomatic acute self-limiting hepatitis or fulminant liver failure with a high rate of mortality, especially in high-risk groups, such as patients with liver-associated comorbidities and immunocompromised or immunosuppressed patients [5,6]. In rare conditions, hepatitis E virus causes chronic infection among immunocompromised people or those with pre-existing liver disease [7]. Patients undergoing regular hemodialysis (HD) are at extreme risk of acquiring bloodstream infections compared to the general population, which may increase proportionally with the duration of hemodialysis; as a result, HEV infection is an important emerging health issue in these patients [3,8]. Despite this importance, HEV is neglected among hemodialysis patients, as they are not routinely screened for HEV, especially in non-endemic countries. To date, numerous studies have investigated the seroprevalence of HEV among hemodialysis patients across the world; however, the available data are conflicting [8,9,10,11,12,13]. Of note, studies using different anti-HEV assays (Abbott GmbH Diagnostica, Wantai HEV IgG ELISA, MP Diagnostics HEV ELISA, Genelabs assay, Fortress Diagnostics, DIA.PRO) differ in their sensitivities for detecting HEV infection [14,15]. A systematic review and meta-analysis compared nine different diagnostic assays for their HEV antibodies detection ability, and revealed remarkable differences regarding their analytical and diagnostic sensitivities [16]. These differences in analytical and diagnostic assay sensitivities necessitate the evaluation of seroconversion panels in addition to end-point titration experiments for an effective assay evaluation. In Greece, the first epidemiological study on hepatitis E in hemodialysis patients was carried out in 1996, which included 420 hemodialysis patients and 316 volunteers. The prevalence rate of hemodialysis patients (6.4%) was not significantly elevated (*p* = 0.07) compared to the group of healthy volunteers (2.2%) [11]. Two more studies of hemodialysis patients showed a prevalence rate of 1.34% in the Epirus region (Western Greece), 9.7% in the region of Agrinio (Central Greece), and 4.8% in Thessaly (Central Greece) [12,13]. As mentioned before, these differences may be attributed to the questionable sensitivity of older ELISA techniques. Therefore, this study aimed to investigate the prevalence and potential risk factors of HEV infection, among hemodialysis patients, by using a modern and reliable ELISA assay. It is an epidemiological study that was based on the detection of anti-HEV IgG antibodies, since our patients had no indicators of acute disease. Additionally, what we were further interested in seeing was whether there was chronic HEV disease in the immunosuppressed group of HD patients, so we examined the samples with a more accurate method, the RT- PCR, in order to identify a possible cure. This is the first report on the prevalence of HEV infection among HD patients in north-eastern Greece.

## 2. Materials and Methods

### 2.1. Patients’ Recruitment and Ethics

The study was conducted from October 2021 to October 2022 and included 405 hemodialysis patients, from six public hemodialysis centers in six cities of the region of Eastern Macedonia and Thrace, in the north-eastern part of Greece. It was the total number of patients monitored in all six public centers of north-eastern Greece during the time period of the study. Hemodialysis centers were chosen according to their geographic location and the socio-professional relations among them. Specifically, 85 subjects were monitored in Alexandroupolis, 60 in Didymoteicho, 62 in Drama, 78 in Kavala, 68 in Komotini, and 52 in Xanthi city. Out of 405 patients, 250 (62.7%) were male and 155 (37.3%) were female. The mean age ± SD of patients was 67 ± 13.6 years with a range of 21–101 years. Our research was in agreement with the ligations of the Bioethics Committee of the University General Hospital of Alexandroupolis, Greece (Ethics Committee 13619). All patients engaged in this study signed the required informed consent. 

### 2.2. Samples’ Processing

All blood samples were centrifuged and the serum was used for downstream applications such as ELISA qualitative HEV IgG detection and quantitative RT-PCR for the inspection of possible active infection. For the RT-PCR application, the HEV RNA extraction was processed by the automated MagCore platform (RBC Bioscience, New Taipei City, Taiwan), protocol code 203. The RNA was extracted from 400 μL of serum samples to 60 μL of elution, and to validate the extraction method 6 μL of armored RNA template was used as Internal Control (IC) for each sample. The RNA extracted was immediately used for downstream applications such as RT-PCR.

### 2.3. Elisa Qualitative Anti-HEV IgG Detection

Freshly collected serum samples were immediately processed with the anti-HEV IgG ELISA kit. All 405 serum samples were treated according to the anti-HEV IgG Elisa kit (Wantai, BioPharm, Beijing, China). The interpretation of the results was in agreement with a cut-off (CO) value equal to 0.08, able to discriminate between anti-HEV IgG-positive samples (CO > A) and anti-HEV IgG-negative (CO < A) individuals. The selection of the anti-HEV IgG ELISA kit Wantai was decided due to the fact that it demonstrated a sensitivity advantage when compared with other commercially available kits. Mansuy et al. studied the sensitivity of the anti-HEV IgG Wantai kit compared with another ELISA kit in an industrialized region in France. In this study, a threefold higher IgG seroprevalence is described, 52.5% versus 16.6%, regarding Wantai and the competitor ELISA kit, respectively [17]. In addition, Bendall et al. found a fourfold higher IgG seropositivity in the UK, 16.2% versus 3.6%, regarding Wantai and the competitor ELISA kit, concluding that other commercially available ELISA anti-HEV IgG kits underestimate the epidemiologic profile of HEV in developed countries [14].

### 2.4. Quantitative RT-PCR for HEV RNA

The quantitative RT-PCR was set by creating a standard curve of 5 differently concentrated HEV RNA quantification standards manufactured according to the 1st WHO International Standard for Hepatitis E Virus RNA Nucleic Acid Amplification Techniques (NAT)-Based Assays (PEI CODE: 6329/10). The Quantitative Standards (QS) had the concentrations QS1 to QS5, 10^4^ to 1 IU/μL accordingly, creating a standard curve with R2 equal to 0.999 and efficiency e = 1.13. The master mix was prepared according to the manufacturer’s instructions (AltoStar^®^HEV RT-PCR kit 1.5 (Altona Diagnostics, Hamburg, Germany)). For the master mix preparation, we mixed the following volumes: Master A 5 μL, Master B 15 μL, and 5 μL of each sample template resulting in a 25 μL total volume. For the validity of the assay, 5 QS (QS1–QS5), 2 non-amplified controls (NAC), and 1 non-template control (NTC) were loaded. The TaqMan RT-PCR assay was performed on the Rotor-Gene Q MDX, 72 rotor platform (Qiagen, Hilden, Germany). The interpretation of the results was completed with the rows that passed the 0.05 threshold, both for the internal control and the HEV RNA detection, showing the extraction validity and the HEV RNA positivity, accordingly. Finally, the formula that was applied to calculate the concentration of HEV RNA-positive samples followed the equation: CT= −3.049 × *log (conc) + 32.238. 

### 2.5. Statistical Analysis

To compare scale (age and duration of hemodialysis) as well as nominal (gender, ethnicity, residence, and occupation) variables between hemodialysis patients with either positive or negative anti-HEV IgG Ab, Student’s t-test and Chi-square test were used, respectively. To explore the independent correlations between anti-HEV IgG Ab status and age, gender, duration of hemodialysis, ethnicity, residence, and occupation, multivariate analyses were performed using binary regression analysis (the probability for stepwise entry and removal were set to 0.05 and 0.10, respectively; the classification cutoff was set to 0.5; and the maximum number of iterations was set to 20). Descriptive statistics are provided either as means along with their relevant standard deviations (SD), or as percentages, for scale and nominal variables, respectively. All reported p-values are two-sided. The level of statistical significance was set to *p* = 0.05. All numerical values have been given with at least two significant digits. The statistical analysis and visualization of results was performed with the use of IBM SPSS Statistics software, version 26.0, for Windows. The MedCalc software Version 20.218 (MedCalc Software Ltd., Ostend, Belgium; 2023) was used to illustrate forest plots. 

## 3. Results

Of the 405 participants, 42 hemodialysis patients (10.4%) were positive for the anti-HEV IgG antibody (10.4% of males and 10.3% of females), but none of them had detectable HEV RNA. More specifically, the six units of north-eastern Greece that participated in this study depicted a range of percentages of anti-HEV IgG-positive subjects, from 3.8% to 21.7%; Alexandroupolis 5.9%, Didymoteicho 21.7%, Drama 16.1%, Kavala 11.5%, Komotini 4.4% and Xanthi 3.8% (Figure 1). Regarding the age distribution, when the patients were divided into five age groups, <40, 40–49, 50–59, 60–69, 70–79 and <80 years old, the highest rate of anti-HEV IgG seroprevalence was observed in those aged 60–69 and 70–79 years, whereas the lowest anti-HEV IgG seropositivity was found in those aged 40–49 years. However, differences in age groups are not significantly correlated with seroprevalence (data not shown). Overall, age does not seem to affect HEV seroprevalence according to univariate (0.641) and multivariate (0.913) analysis (Table 1). According to the place of residency, hemodialysis patients resident in Didymoteicho and Drama showed the highest seroprevalence rates (21.7% and 16.1%, respectively). Grouping the cities in wider areas, it can be clearly seen that the region of Evros (Didymoteicho and Alexandroupolis) and the provinces of Kavala and Drama showed a statistically significant difference in terms of seropositivity compared to the provinces of Komotini and Xanthi (*p* = 0.028). Regarding religion, HEV seroprevalence among Christian hemodialysis patients (36 out of 327, 11%) in comparison with Muslims (6 out of 78, 7.7%) was higher, albeit not significant (*p* = 0.388). In addition, no significant association was observed between HEV seroprevalence and the gender distribution of hemodialysis patients. HEV seroprevalence among hemodialysis patients was significantly associated with their occupation working with livestock or agriculture. Hemodialysis patients who were engaged in animal husbandry, hunting, pig slaughtering, or farming in general, had significantly higher anti-HEV IgG prevalence than those who were not (*p* = 0.006). Additionally, we found that Evros (45/145; 31%) and Kavala/Drama (44/140; 31.4%) had a significantly higher number of patients in agricultural or livestock occupations than Komotini/Xanthi (14/120; 11.7%) (*p* = 0.0002 and 0.0001, respectively). All hemodialysis patients had normal levels of liver enzymes; therefore, there was no significant difference between seronegative and seropositive hemodialysis patients regarding levels of AST and ALT. All anti-HEV seropositive samples were negative for HEV RNA. Τhe socio-demographic characteristics and quantitative variables of hemodialysis patients are presented in Table 1.

Binary logistic regression analysis confirmed the independent correlation of HEV seroprevalence with residence, as well as livestock- or agriculture-based occupation. In contrast, neither the duration of hemodialysis nor religion, age, or gender were associated with the seroprevalence of HEV infection among hemodialysis patients (Figure 2).

## 4. Discussion

In recent years, a contemporary emergence of HEV infection has been observed in developed countries. Developing data on HD patients is considered of major importance, in relation to considerations of candidacy for transplantation, and since a significant percentage of viremic transplant patients could develop chronic hepatitis and cirrhosis as a result of their immunocompromised condition [18]. Our study showed an overall anti-HEV IgG prevalence of 10.4%, suggesting a high degree of previous exposure to HEV. Additionally, there was a significant variation in HEV IgG prevalence between HD centers ranging from 3.8% to 21.7%, reflecting geographical differences.

A similar high seroprevalence has been shown in different studies of HD patients around the world. In particular, a Croatian study revealed 27.9% of seropositive individuals under HD, with none of them, however, presenting positivity for HEV RNA. Although this seropositivity seems to be higher than the result in our findings, the absence of viremia agrees with our data [18]. Moreover, the discrepancy in seropositivity rates may be explained by the different diagnostic techniques used; an anti-HEV IgG Elisa kit (Wantai, BioPharm, Beijing, China) was used in our study, while an anti-Hepatitis E virus ELISA (Euroimmun, Lübeck, Germany) was used in the Croatian study. Such differences in seroprevalence were noted in many studies, e.g., in a Turkish study [19], both south-eastern (20–23%) and northern (10–16%) parts of the country showed significantly higher seroprevalence of HEV infection than in our study, while in a Swedish study including 182 HD patients, seropositivity was lower (6%) [9].

In addition, our study revealed important variability in IgG seropositivity among the different areas studied. This difference seems to have a major common denominator, namely, lifestyle habits. More specifically, in Kavala/Drama (19/140; 13.6%) and Evros (18/145; 12.4%) provinces, significantly higher seroprevalence (*p* = 0.028) was found in comparison with Komotini/Xanthi (5 out of 120, 4.2%) province, which is probably closely related to the significantly higher rates of agriculture- or livestock-based occupations in these areas, as was described in the results. In particular, residents of these regions hunt wild animals for meat. As previously reported, the consumption of wild boar has been associated with acute hepatitis E in many European countries, such as Germany [20]. Additionally, in a study from Poland, raising and having contact with farm animals was associated with HEV infection among hunters [21]. Thus, several studies have proven that people working in agriculture or who are exposed to specific animals (deposits of HEV), such as pigs, wild boars and deers, have a greater risk of HEV infection [22,23], underlining the zoonotic risk of the infection in most European studies [24,25,26].

Religion is another aspect that may affect seropositivity. The correlation between seropositivity and religious beliefs has been recognized through a precise meta-analysis, wherein European countries such as France (10.8%), Germany (3.0%), Italy (2.9%), Spain (2.0%) and Greece (6.4%) were shown to have a mean HEV IgG prevalence of 5.02 ± 0.04%, while non-European countries such as India (40%), Taiwan (31%), Turkey (20%), Iran (13.3%) and Egypt (22.9%) showed a significantly high seroprevalence mean value equal to 25 ± 0.1% [8,19,27]. Given the fact that Christianity dominates in European countries and Islamism dominates in countries such as Turkey, Egypt and Iran, we would expect a lower seroprevalence in Muslim countries due to the religious restriction of pork consumption. However, this wide HEV seroprevalence range reflects the differences in viral circulation within different geographical regions, where the distribution of genotypes varies. It is worth noticing that genotypes are not closely described in the studies, leaving the question of genotype-dependent seropositivity open. Moreover, the consumption of different animal deposits (deer) may also affect seropositivity as well. Although Muslim patients are underrepresented in our study, we did not find any significant differences between the two religious groups in our province (Table 1). The above findings suggest the possibility that husbandry work, and not religious beliefs and practices, represents the most precipitating factor in relation to contracting HEV infection.

In our study, age and age distribution did not seem to affect HEV seropositivity (*p* = 0.913), although a tendency has been found. In contrast, the study of Jelicic et al. [28] showed a significant increase in seropositivity with age. In particular, patients aged less than 30 years had a higher rate of seropositivity in comparison with those older than 60 years (*p* < 0.001). Additionally, the Croatian study previously reported showed that older age was associated with higher seroprevalence, suggesting a lifelong cumulative exposure to the virus [18]. Similarly, a correlation of age and seropositivity was observed by Ouji et al. [29], who studied seroprevalence in dialysis patients in Iran and revealed that the mean age of HEV-seropositive patients was significantly higher than that of seronegative ones. Lastly, a Swedish study by Sylvan et al. [9] noted that in persons younger than 40 years, the percentage of seropositive individuals was lower than in patients older than 40 years (*p* < 0.05).

Furthermore, as reported by previous studies [19,28,29], factors such as gender and hemodialysis duration do not seem to have a statistically significant effect on HEV seroprevalence in HD patients (Table 1). We note that there was no significant difference in HD duration (measured in years) among patients from each region (Komotini/Xanthi 5.47 +/− 3.26, Kavala/Drama 6.01 +/− 3.57, Evros 5.73 +/− 3.79; ANOVA *p* = 0.481).

Finally, it is important to note that this study has certain limitations. Monitoring anti-HEV IgG seroprevalence is essential when determining the presence of past infection; nevertheless, patients on hemodialysis may be immunocompromised, and the results of any immunoglobulin-based assay in this context may underestimate true prevalence. Additionally, although the duration of HD does not appear to have been a significant risk factor for HEV infection in the present study, HD alone remains a critical risk factor for HEV infection due to the continuous transfusions undergone by immunocompromised HD patients. Despite these limitations, this is the first study in north-eastern Greece to examine the incidence of anti-HEV among HD patients, providing new information related to HEV epidemiology in this region.

## 5. Conclusions

In conclusion, our study revealed a high seroprevalence of HEV infection among HD patients, making routine screening in this setting imperative. Moreover, living in rural parts of the country and having an agriculture- and livestock-based occupation represent significant risk factors for HEV seropositivity. However, although the zoonotic transmission route seems to be the most important, the whole range of risk factors for HEV acquisition in HD patients is yet to be elucidated.

## Figures and Tables

**Figure 1 pathogens-12-00667-f001:**
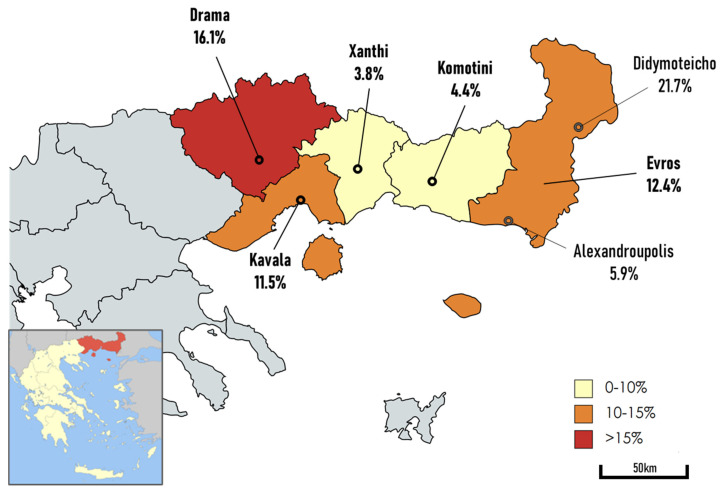
Hepatitis E seroprevalence in hemodialysis units according to geographic location of north-eastern Greece.

**Figure 2 pathogens-12-00667-f002:**
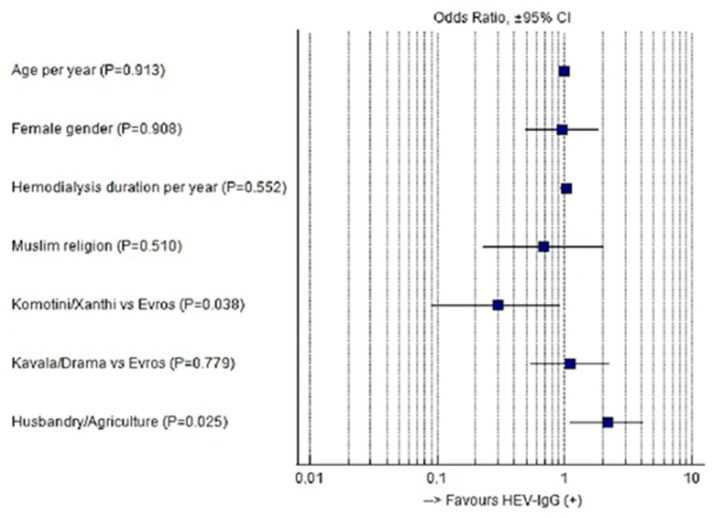
Forest plot depicting OR ± 95% CI values as derived from multivariate analysis given in Table 1.

**Table 1 pathogens-12-00667-t001:** Characteristics of hemodialysis patients and univariate as well as multivariate analysis; OR > 1 indicates unfavorable effect, while OR < 1 is a favorable effect.

Parameters	Mean (SD) †, N (%) ‡	Anti-HEV IgG (+); n = 42 (%)	Anti-HEV IgG (−); n = 363 (%)	Univariate Analysis *p*-Value	Multivariate Analysis OR; ±95%CI ⁋	Multivariate Analysis *p*-Value ⁋
**Age**						
Mean (SD)	67.1 ± 13.6	66.1 ± 11.9	67.2 ± 13.8	0.641	0.999 (0.974–1.023)	0.913
**Gender**						
Males	250 (62.7)	26 (61.9)	224 (61.7)	0.980	1.000	
Females	155 (37.3)	16 (38.1)	139 (38.3)		0.961 (0.490–1.883)	0.908
**Hemodialysis duration ***						
Mean (SD)	5.75 ± 3.56	6.14 ± 3.66	5.71 ± 3.56	0.452	1.027 (0.941–1.121)	0.552
**Religion**						
Christians	327 (80.7)	36 (85.7)	291 (80.2)	0.388	1.000	
Muslims	78 (19.3)	6 (14.3)	72 (19.8)		0.694 (0.234–2.059)	0.510
**Residence**						
Evros (Alexandroupolis/Didymoteicho)	145 (35.8)	18 (42.9)	127 (35.0)	0.028	1.000	
Komotini/Xanthi	120 (29.6)	5 (11.9)	115 (31.7)		0.296 (0.094–0.936)	0.038
Kavala/Drama	140 (34.6)	19 (45.2)	121 (33.3)		1.107 (0.543–2.260)	0.779
**Husbandry/Agriculture**						
No	302 (74.6)	24 (57.1)	278 (76.6)	0.006	1.000	
Yes	103 (25.4)	18 (42.9)	85 (23.4)		2.155 (1.102–4.214)	0.025

**Abbreviations**: SD: standard deviation; OR: odds ratio; CI: confidence interval. † For scale variables. ‡ For nominal variables. ⁋ Binary regression model (Omnibus test for model coefficients P: 0.052; Nagelkerke R2: 0.070; Hosmer and Lemeshow test P: 0.615). * Measured in years.

## Data Availability

Data is unavailable due to privacy and ethical restrictions.

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
