# Peer review of "Hepatitis E Virus (HEV) Infection in Hemodialysis Patients: A Multicenter Epidemiological Cohort Study in North-Eastern Greece"

_pathogens, 2023, doi:10.3390/pathogens12050667_

Round 1

Reviewer 1 Report

Kogias et al., performed a seroprevalence study on HEV infection in hemodialysis patients in Greece.

I have a couple of comments:

-The patients' recruitment needs to be re-written: what were the inclusion/exclusion criteria? What was the time period considered? What was the location from where they were enrolled (i.e. hospital, center...)? Please refer to the cities describing where they are located, it is not easy if you don't know the regional Greek geography. There was any specific reason for the different enrolled subjects in each city. Why these cities were chosen?

-Please correct Table 1, figure 1 is incorporated into

-Please mark with bold or anything else the significant variables in Figure 2.

-Please correct line 209. The Croatian seropositivity is lower than the one observed in this study.

-Please describe better the limitations of this study.

Reviewer 2 Report

Kogias et al., in the article “Hepatitis E virus (HEV) infection in Hemodialysis Patients: A Multicenter Epidemiological Cohort Study in North-Eastern Greece” investigated for the first time in this region the seroprevalence for HEV in 405 hemodialysis patients, analysing some risk factor that could be associated with HEV infection.

In the Abstract “Hepatitis E virus” should be added before HEV, together with an introductive general sentence for this virus.

The Introduction section should be refined. Particularly, in the first sentences the Authors should add major details about the classification of the HEV genotypes. Regarding lines 55-58, it is not clear that HEV3 and HEV4 are zoonotic and that animals are asymptomatic but in humans the infection could cause hepatitis E. Moreover, lines 60-63 need to be rephased. In fact, the Authors did not mention the asymptomatic infection in humans and “pregnant women” should be deleted because only HEV1 infections could cause a high mortality rate during pregnancy.

The list in lines 72-73 should be completed or deleted.

In line 89, which region are the Authors referring to?

All over the Material and Methods section, “u” has to be changed in “μ”.

In lines 119 did the Authors mean 104?

In the Results section age of seropositive patients is introduced for the first time. Please add in the Materials and Methods section more details on the age stratification of the population analysed.

Can the Authors discuss the decision to perform their seroprevalence survey with anti-HEV IgG ELISA kit by Wantai? Which criteria did they apply to select this kit?

In the Results section, please round up 10.37% to 10.4%.

In the PDF version the table is “broken” by the Figure 1. However, please add that numbers within baskets are percentages. What unit of measurement was used for the hemodialysis duration?

The sentences in lines 175-177 should be rephrased because they are repetitive.

In Discussion section, please add references in lines 198-199 and 199-202.

In line 261, given the results illustrated in lines 218-231, why did the authors list the place of residence with the factor that not are associated with a higher HEV infection?

Can the Authors discuss because, in their opinion, seroprevalence for HEV were higher in patients aged 21-39, decreased in 40-49 and come back higher in patients aged 60-69? What about the other age group?

Why was not an investigation aimed to detected anti-IgM performed?

In Greece, is there any data available about the seroprevalence on general population to compare with that obtained in haemodialysis patients?

Reviewer 3 Report

The authors measured seroprevalence of anti-HEV IgG antibodies (Wantai) among hemodialysis patients of North-Eastern Greece (6 centers from 4 region).

Totally 42 out of 405 hemodialysis patients were positive for anti-HEV IgG (10.37%), but none of them had detectable HEV RNA.

The authors concluded that HEV seropositivity among hemodialysis patients was significantly associated with area of residence and agricultural or livestock occupation, and that no association was found with religion, gender distribution and hemodialysis duration.

There are some major problems.

1.     The authors reported high seroprevalence of anti-HEV IgG antibodies among hemodialysis patients. It seems that the authors think that hemodialysis is one of risk factors of HEV infection. However, the authors concluded that no association was found with hemodialysis duration. Do the authors think that hemodialysis itself is associated with HEV infection not hemodialysis duration? The authors should carefully answer the question and must state the reasons or discussions enough in detail. Is the duration of hemodialysis the only cue or key for investigating the association of HEV infection and hemodialysis?

2.     Page 4 line 169 The authors wrote:

 “HEV seroprevalence among hemodialysis patients was significantly associated with their livestock or agricultural occupation.”

Page 7 line 218 The authors also wrote:

“In addition, our study revealed important variability of IgG seropositivity among the different areas studied. This difference seems to have a major common denominator, namely lifestyle habits. More specifically, in Kavala/Drama (19 out of 140, 13.6%) and Evros (18 out of 145, 12.4%) provinces significantly higher seroprevalence was found in comparison with Komotini/Xanthi (5 out of 120, 4.2%) province, which possibly reflects the more extensive agricultural or livestock occupation in these areas.”

If the authors say so, please indicate difference of No. of patients who were working on agricultural or livestock occupation in these different areas. Are those differences significant by P values?

3.     There are differences of seroprevalence among regions. The authors concluded that no association was found with hemodialysis duration. The authors should indicate difference of hemodialysis duration among patients of each region. Are those differences significant by P values?

Minor problems

1.     Page 4 line 149  “Out of 405 hemodialysis patients, 250 (62.7%) were male and 155 (37.3%) were female. The mean age ± SD of hemodialysis patients was 67±13.6 years with a range of 21-101 years. “This description should be move to Materials and Methods section.

2.     The authors had better add scale of distance to figure 1.

3.     Page 5-6, Table 1 and Figure 1

Table 1 is divided by Figure 1 in this draft.

Round 2

Reviewer 3 Report

The manuscript was adequately improved.